# OpenReview forum: "Untargeted Jailbreak Attack"
_ICLR.cc/2026/Conference — ICLR 2026 Conference Withdrawn Submission_

### Official Review · Reviewer_EaDK · 2025-10-30

**Soundness:** 4
**Presentation:** 3
**Contribution:** 4
**Rating:** 8
**Confidence:** 4

**Summary:**

This paper introduces the Untargeted Jailbreak Attack (UJA), a gradient-based method for jailbreaking Large Language Models (LLMs). Existing attacks, such as GCG and COLD-Attack, are described as "targeted", meaning they optimize an adversarial prompt to align the LLM's output with a predefined target response. The paper states this approach constrains the optimization search space.

UJA, in contrast, is an untargeted attack. Its objective is to maximize the unsafety probability of the LLM response without enforcing a predefined pattern. This unsafety probability is quantified using a judge model.

Because this objective is non-differentiable, the paper proposes a two-stage optimization process:
1.  Stage 1 (Adversarial Response Optimization): An optimal harmful response is identified by optimizing over the judge model's representations.
2.  Stage 2 (Adversarial Prompt Optimization): An adversarial prompt is optimized to make the target LLM generate the response found in Stage 1. This stage uses a "gradient projection" technique to transfer information between the different token spaces of the judge model and the target LLM.

The paper's contributions include the formulation of this untargeted objective and the two-stage optimization method. Evaluations in the paper show UJA achieves over 80% Attack Success Rate (ASR) with 100 optimization iterations, which is reported as outperforming baselines by over 20%.

**Strengths:**

Originality: The work's originality lies in its critique of existing "targeted" jailbreak attacks and its novel formulation of an "untargeted" objective. This new problem definition, which maximizes unsafety probability, presents a distinct conceptual approach to gradient-based attacks.

Clarity: The paper is well-structured. It clearly articulates the limitations of prior work and logically presents its proposed two-stage methodology. The authors identify the non-differentiable nature of their objective and detail their proposed solution.

Quality: The methodological quality is high. The paper provides a theoretical justification for its decomposition of the non-differentiable objective. This rigor extends to the empirical evaluation, which includes comprehensive ablation studies that proactively address key questions about the framework's components, such as the necessity of the response optimization stage and the impact of different judge models.

Significance: The paper's significance is demonstrated by its strong empirical results. The proposed method achieves a substantially higher attack success rate on benchmarks compared to existing state-of-the-art methods, establishing a new and more effective baseline for jailbreak attacks.

**Weaknesses:**

The paper's primary weakness lies in its "untargeted" claim. The methodology is more accurately described as a dynamic target-finding attack. The process is not target-free; Stage 1 is explicitly designed to find a single, optimal target response, and Stage 2 then optimizes the prompt to match that specific target. This is still a form of targeted optimization, which re-introduces the kind of constraint the paper claims to overcome.

A second, and more fundamental, limitation is the framework's critical dependency on the judge model. As suggested by the ablation study on different judges, the attack's effectiveness is fundamentally capped by the classification performance and biases of the chosen judge model. The method is optimized to generate a response that fools that specific judge, which may not be the same as generating a response that is universally or practically harmful. The attack's success is relative to its own component, meaning it may simply be overfitting to the judge's specific vulnerabilities.

Finally, the two-stage, two-model architecture introduces significant practical complexity. It requires white-box gradient access to both the target LLM and the judge model, plus a "gradient projection" step to bridge their vocabularies. This is a far more complex setup than single-model baselines. The paper's focus on efficiency in iterations may obscure the true computational cost and setup overhead of this more intricate optimization.

**Questions:**

The "untargeted" claim is confusing. The method finds a specific target response in Stage 1 and then optimizes the prompt to match that target in Stage 2. Could the authors clarify how this is fundamentally different from a targeted attack, other than the target being generated dynamically?

The ablation study on judge models is a key part of the paper. Were any other judge models used in the experiments besides GPTFuzzer and Llama-Guard-3?

The paper highlights efficiency in terms of iterations. However, the two-stage, two-model approach seems computationally complex. Could the authors provide a comparison of the total wall-clock time and memory usage required for UJA versus the baselines (e.g., GCG, COLD-Attack) to achieve their reported results?

---

> ### Author Response · Authors · 2025-11-26
> **Reply to Reviewer EaDK**
>
> >**Q1. The "untargeted" claim is confusing. The method finds a specific target response in Stage 1 and then optimizes the prompt to match that target in Stage 2. Could the authors clarify how this is fundamentally different from a targeted attack, other than the target being generated dynamically?**
>
> Thanks for raising this question. In this paper, we formulate the attack problem as Eq.1 and Eq.2, which aim to maximize the unsafety probability without specifying any target response. From our perspective, Stage 1 and 2 present an approximation method to solve Eq.1 and Eq.2, and Eq.1 and 2 are the formal formulation of this paper. Therefore, we call our attack “untargeted attack”, which means the problem formulation Eq.1 and 2 do not have any target response.
>
> >**Q2. The ablation study on judge models is a key part of the paper. Were any other judge models used in the experiments besides GPTFuzzer and Llama-Guard-3?**
>
> We use Qwen3Guard-Gen-0.6B to provide gradient feedback during optimization and report the results in the following table.
>
> | Metric | GPTFuzz | Qwen3Guard-Gen-0.6B |
> |:------:|:-------:|:------------------:|
> | ASR-G (%) | 89.0      | 76.0       |
> | ASR-H (%) | 67.0      |  58.0    |
>
> While all judges enable successful optimization, GPTFuzzer achieves the highest ASR-G, demonstrating its superior capability in providing stable gradient feedback for prompt refinement. This performance advantage motivated our selection of GPTFuzzer as the primary judge in the main experiments.
>
> >**Q3. The paper highlights efficiency in terms of iterations. However, the two-stage, two-model approach seems computationally complex. Could the authors provide a comparison of the total wall-clock time and memory usage required for UJA versus the baselines (e.g., GCG, COLD-Attack) to achieve their reported results?**
>
> We have evaluated all baselines under 100 iterations and report the overhead results in the following table.
>
> | Method       | ASR-G (%) | Time (h) | GPU Cost ($) | Cost per success ($)|
> |:-------------|:------:|:--------:|:---------:|:-----------------:|
> | GCG          | 50.0  | 42.8     | 19.26     | 0.39              |
> | COLD         | 52.0  | 0.4      | 0.18      | 0.003             |
> | DRL          | 30.0  | 0.08     | 0.04      | 0.001             |
> | PAP          | 21.0  | 0.87     | 0.39      | 0.02              |
> | AdvPrefix    | 40.0  | 45.6     | 20.52     | 0.51              |
> | I-GCG        | 23.0  | 2.52     | 1.13      | 0.05              |
> | Llm-adaptive | 51.0  | 0.8      | 0.85      | 0.38              |
> | Ours (UJA)   | 89.0  | 5.42     | 2.4       | 0.03              |
>
> ·GPU cost estimated using A6000 pricing on Vast.ai: $0.45/hour.
>
> These results demonstrate that UJA achieves the highest attack success rate while maintaining competitive computational efficiency. With a cost per successful attack of only $0.03, our method proves to be significantly more cost-effective than most baselines, particularly those with comparable ASR performance.

---

### Official Review · Reviewer_ZE6f · 2025-10-31

**Soundness:** 3
**Presentation:** 3
**Contribution:** 3
**Rating:** 6
**Confidence:** 3

**Summary:**

This paper proposes Untargeted Jailbreak Attack (UJA), a gradient-based jailbreak method that does not require predefined target responses. Unlike methods like GCG and COLD-Attack that optimize for predefined prefixes, UJA maximizes the unsafety probability of LLM responses via a judge model. It decomposes the non-differentiable objective into two differentiable sub-objectives: optimizing an optimal harmful response and the corresponding adversarial prompt. Extensive experiments on 8 LLMs (e.g., Llama-3, Qwen-2.5) and 2 datasets (AdvBench, HarmBench) show UJA achieves over 80% attack success rate (ASR). It also exhibits strong transferability and robustness against defenses like Perplexity and SmoothLLM.

**Strengths:**

1. The paper tackles an important problem inherent in existing gradient-based jailbreak methods: their reliance on predefined affirmative response prefixes (e.g., "Sure, here is...") as optimization targets. This dependency requires prior knowledge of model-specific response patterns and can lead to optimization futility when the chosen prefix poorly aligns with the target LLM's natural output distribution.

2.  The consistent superiority over seven baseline methods provides strong empirical support.

3. UJA overcomes a fundamental challenge in transferring optimization signals across the target model and judge model with different tokenization schemes.

**Weaknesses:**

1. In the experimental section, the paper only uses GPTFuzzer as the judge model to provide feedback, which raises concerns about the reliability of the proposed method.

2. The method exhibits high dependence on the judge model, often requiring a specially fine-tuned judge model. This actually imposes constraints on the method’s practicality. In new malicious scenarios (e.g., biosecurity scenarios), these fine-tuned judge models lack relevant training and may need to be re-fine-tuned. The authors should objectively examine the potential limitations of the method.

3. Lacks discussion on some existing works. For instance, there are black-box jailbreaking suffix optimization works that also do not require targets and share the same idea of using a judge model to provide feedback signals (e.g., [1] An Optimizable Suffix Is Worth A Thousand Templates: Efficient Black-box Jailbreaking without Affirmative Phrases via LLM as Optimizer; [2] Jailbreaking Black Box Large Language Models in Twenty Queries). The authors should discuss them in the paper.

**Questions:**

1. Have the authors considered testing the method’s performance when using other judge models for feedback?

---

> ### Author Response · Authors · 2025-11-26
> **Reply to Reviewer ZE6f**
>
> >**Q1: Have the authors considered testing the method’s performance when using other judge models for feedback?**
>
> We use Qwen3Guard-Gen-0.6B to provide gradient feedback during optimization and report the results in the following table.
>
> | Metric | GPTFuzz | Qwen3Guard-Gen-0.6B |
> |:------:|:-------:|:------------------:|
> | ASR-G (%) | 89.0      | 76.0       |
> | ASR-H (%) | 67.0      |  58.0    |
>
> While all judges enable successful optimization, GPTFuzzer achieves the highest ASR-G, demonstrating its superior capability in providing stable gradient feedback for prompt refinement. This performance advantage motivated our selection of GPTFuzzer as the primary judge in the main experiments.
>
> >**W1. In the experimental section, the paper only uses GPTFuzzer as the judge model to provide feedback, which raises concerns about the reliability of the proposed method.**
>
> Please refer to Q1 Answer.
>
> >**W2. The method exhibits high dependence on the judge model, often requiring a specially fine-tuned judge model. This actually imposes constraints on the method’s practicality. In new malicious scenarios (e.g., biosecurity scenarios), these fine-tuned judge models lack relevant training and may need to be re-fine-tuned. The authors should objectively examine the potential limitations of the method.**
>
> We focus on maximizing the unsafety probability of the LLM output, which can be approximated by a judge scoring model. Although we use one judge to approximate this probability during optimization, the resulting prompts remain effective across different judges and evaluators.
> As shown in Table 1 in the paper, we use GPTFuzzer to optimize the prompts and another widely-used evaluator (HarmBench-Llama-2-13b-cls) for evaluation. Table 1 (in the paper) shows that UJA achieves higher ASRs than other baselinese.
>
> >**W3. Lacks discussion on some existing works. For instance, there are black-box jailbreaking suffix optimization works that also do not require targets and share the same idea of using a judge model to provide feedback signals (e.g., [1] An Optimizable Suffix Is Worth A Thousand Templates: Efficient Black-box Jailbreaking without Affirmative Phrases via LLM as Optimizer; [2] Jailbreaking Black Box Large Language Models in Twenty Queries). The authors should discuss them in the paper.**
>
> While [1] and [2] also rely on judge models in a black-box setting, UJA operates in a white-box setting where gradients are directly accessible. We compare UJA with [1] and [2] under a strict 100-iteration budget and report the results in the following table.
>
> |                  | Llama-3        |           |           | Qwen-2.5       |           |           |
> |--------------|----------------|-----------|-----------|----------------|-----------|-----------|
> |    | [1] ECLIPSE    | [2] PAIR  | UJA       | [1] ECLIPSE    | [2] PAIR  | UJA       |
> | ASR-G (%)    | 64.0           | 59.0      | 89.0      | 61.0           | 34.0      | 74.0      |
> | ASR-H (%)    | 53.0           | 66.0      | 67.0      | 50.0           | 38.0      | 55.0      |
>
> The results show that UJA achieves higher attack success rates under the same iteration constraint. We will include a detailed discussion and comparison with these methods in the revised manuscript.

---

### Official Review · Reviewer_ogst · 2025-10-31

**Soundness:** 3
**Presentation:** 2
**Contribution:** 2
**Rating:** 4
**Confidence:** 4

**Summary:**

This paper proposes UJA (Untargeted Jailbreak Attack) — a novel gradient-based jailbreak framework that, unlike prior targeted methods (e.g., GCG, COLD), removes the dependence on predefined target prefixes.
UJA formulates jailbreak as maximizing the unsafe probability of LLM outputs judged by an external classifier

**Strengths:**

1. Experiments cover several models and datasets, showing reasonably consistent improvements in ASR over certain baselines.

2. The proposed method shows strong transferability across architectures, with improved efficiency (fewer iterations, lower GPU cost).

**Weaknesses:**

- Dependence on the judge model.
The method heavily relies on a differentiable judge model $J(\cdot)$ to compute gradients of “unsafe probability.” In practice, such judges encode alignment criteria that are usually proprietary or confidential. Using them for optimization exposes sensitive alignment information and might not reflect a realistic attacker setting.

- Insufficient methodological clarity.
The paper does not specify critical implementation details.  For example, it remains unclear:
  - What exactly is z? Is it a pooled embedding, token-level hidden representation, or logits vector?
  - How to obtain the final prompt $p$ based on $z$

- Incomplete baseline comparison.
The experimental section omits several relevant and recent jailbreak approaches such as $\texttt{AdvPrompter}$ and R1-style reinforcement-based methods.

**Questions:**

Q1. In Table 3, models such as Mistral and Vicuna appear to have weak defense capability—they tend to produce unsafe outputs even when conditioned with compliance instructions. However, the baseline $\texttt{llm-adaptive}$ achieves unusually low ASR despite being a context-adaptive method.  Can the authors explain why $\texttt{llm-adaptive}$ performs so poorly under these conditions?

Q2. The optimization process of UJA essentially maximizes a scalar unsafe score from a judge model, which resembles a reinforcement learning setup (reward maximization). Why a purely gradient-based approach was preferred, and what advantages or limitations it brings compared to an RL formulation.

**Details Of Ethics Concerns:**

The posted examples of harmful responses may contain harmful information.

---

> ### Author Response · Authors · 2025-11-26
> **Reply to Reviewer ogst**
>
> >**Q1. In Table 3, models such as Mistral and Vicuna appear to have weak defense capability—they tend to produce unsafe outputs even when conditioned with compliance instructions. However, the baseline llm-Adaptive achieves unusually low ASR despite being a context-adaptive method. Can the authors explain why llm-Adaptive performs so poorly under these conditions?**
>
> Under the consistent 100-iteration budget used in our evaluation, both I-GCG and LLM-Adaptive exhibit significantly reduced ASR compared to their original reported performance. This is because these methods are designed for and typically require thousands of iterations (LLM-Adaptive originally uses up to 10,000 steps) to converge effectively. The constrained optimization budget therefore leads to the substantial performance degradation observed in our evaluation.
>
> >**Q2. The optimization process of UJA essentially maximizes a scalar unsafe score from a judge model, which resembles a reinforcement learning setup (reward maximization). Why a purely gradient-based approach was preferred, and what advantages or limitations it brings compared to an RL formulation.**
>
> UJA employs a gradient-based approach rather than an RL formulation to directly leverage the differentiability of the judge model. This provides precise, dense gradient signals for token-level updates, in contrast to the sparse and high-variance rewards typical in RL, leading to more stable and efficient optimization.
>
> >**W1. Dependence on the judge model. The method heavily relies on a differentiable judge model to compute gradients of “unsafe probability.” In practice, such judges encode alignment criteria that are usually proprietary or confidential. Using them for optimization exposes sensitive alignment information and might not reflect a realistic attacker setting.**
>
> We focus on maximizing the unsafety probability of the LLM output, which can be approximated by a judge scoring model. Although we use one judge to approximate this probability during optimization, the resulting prompts remain effective across different judges and evaluators.
> First of all, as shown in Table 1 in the paper, we use GPTFuzzer to optimize the prompts and another widely-used evaluator (HarmBench-Llama-2-13b-cls) for evaluation. Table 1 shows that UJA achieves higher ASRs than other baselinese.
> Moreover, we further use GPTFuzzer to optimize the prompts and GPT-4o as the evalutor and report the results in the following table.
>
> | Method      | GCG | Cold-attack | DRL | PAP | AdvPrefix | I-GCG | Adaptive | UJA |
> |:-----------:|:---:|:-----------:|:---:|:---:|:---------:|:-----:|:---------:|:---:|
> | ASR (%)     | 50.0% | 53.0% | 44.0% | 31.0% | 26.0% | 14.0% | 13.0% | 72.0% |
>
> The results demonstrate that UJA still consistently outperforms other baselines. This indicates that UJA makes the model genuinely more prone to unsafe responses rather than merely deceive the judge scoring model.
>
> >**W2. Insufficient methodological clarity. The paper does not specify critical implementation details. For example, it remains unclear: What exactly is z? Is it a pooled embedding, token-level hidden representation, or logits vector? How to obtain the final prompt p based on z**
>
> In this paper, \(z \) denotes the logits sequence.
>
> We follow the decoding procedure of COLD-Attack. After obtaining a sampled logit sequence \( z \), we decode it into a prompt \( p \).
> Specifically, for each position \( i \), we first use the underlying LLM to compute the next-token distribution conditioned on previously decoded tokens:$p_{\text{LM}}(v \mid \mathbf{x} \oplus p_{\lt i})$
>
> For each position \( i \), we construct the top-\(k\) candidate set \( \mathcal{V}\_{i}^{k} \) from this distribution, and then greedily select the token with the highest logit value within this set:
>
> $$
> p_i = \arg\max_{v \in \mathcal{V}\_{i}^{k}} z_i(v)
> $$
>
> Finally, UJA outputs the decoded text sequence$p = (p_1, \ldots, p_L)$
> which is generally fluent because each token is guided by the LLM’s next-token distribution.
>
> >**W3. Incomplete baseline comparison. The experimental section omits several relevant and recent jailbreak approaches such as AdvPrompter and R1-style reinforcement-based methods.**
>
> We have now included comparisons with AdvPrompter and the reinforcement-based method (DRL) under identical evaluation settings (100 iterations), and report the results in the following table.
>
> |              | Llama-3|       |             | Qwen-2.5|       |               |
> |--------------|:---------------:|:-----:|:-----------:|:----------------:|:-----:|:-------------:|
> |    | AdvPrompter | DRL  | UJA  | AdvPrompter | DRL  | UJA  |
> | ASR-G (%)  | 50.0 | 30.0 | 89.0 | 34.0 | 36.0 | 74.0 |
> | ASR-H (%) | 58.0 | 28.0 | 67.0 | 35.0 | 64.0 | 55.0 |
>
> The results demonstrate that UJA consistently outperforms these recent baselines across both evaluation metrics. We will include these baselines in the revised manuscript to provide a more comprehensive comparison.

---

### Official Review · Reviewer_rB23 · 2025-10-31

**Soundness:** 1
**Presentation:** 1
**Contribution:** 1
**Rating:** 0
**Confidence:** 3

**Summary:**

The paper attempts to address limitations in existing 'targeted' gradient-based jailbreak attacks, which aim to force a model to output a predefined phrase, constraining effectiveness and efficiency. It proposes the Untargeted Jailbreak Attack (UJA), which instead seeks to elicit any unsafe response, thereby broadening the potential adversarial search space. It is implemented by maximizing the unsafe score of the target LLM's response, obtained from a judge model.

**Strengths:**

The paper identifies a genuine problem clearly. Nevertheless, it is flawed in other aspects (see "Weaknesses").

**Weaknesses:**

● How does the proposed algorithm ensure that a response is genuinely unsafe, rather than merely deceiving the employed judge model?

● The writing strongly requires enhancement, including but not limited to streamlining verbose descriptions and ensuring the correct use of mathematical notation.

● Line 301: How is it guaranteed that a valid candidate jailbreak prompt is obtained?

● Regarding the experiments: Why does I-GCG perform worse than GCG? Furthermore, why does LLM-Adaptive perform poorly when attacking Vicuna and Mistral? In its original paper, the success rate against these two models was close to 100%.

**Questions:**

See Weaknesses.

---

> ### Author Response · Authors · 2025-11-19
> **Reply to Reviewer rB23**
>
> **W1**. How does the proposed algorithm ensure that a response is genuinely unsafe, rather than merely deceiving the employed judge model?
>
> We focus on maximizing the unsafety probability of the LLM output, which can be approximated by a judge scoring model. Although we use one judge to approximate this probability during optimization, the resulting prompts remain effective across different judges and evaluators.
>
> First of all, as shown in Table 1 in the paper, we use GPTFuzzer to optimize the prompts and another widely-used evaluator (HarmBench-Llama-2-13b-cls) for evaluation. Table 1 (in the paper) shows that UJA achieves higher ASRs than other baselinese.
>
> Moreover, we further use GPTFuzzer to optimize the prompts and GPT-4o as the evalutor and report the results in the following table.
>
>
> | Method       | GCG | Cold | DRL | PAP | AdvPrefix | I-GCG | llm-adaptive | UJA (Ours) |
> |:------------:|:---:|:----:|:---:|:---:|:---------:|:-----:|:--------:|:----------:|
> | ASR (%)      | 50.0| 53.0 |44.0 |31.0 |   26.0    | 14.0  |  13.0    |   72.0     |
>
>
>
>
> The results demonstrate that UJA still consistently outperforms other baselines. This indicates that UJA makes the model genuinely more prone to unsafe responses rather than merely deceive the judge scoring model.
>
>
> **W2**. Line 301: How is it guaranteed that a valid candidate jailbreak prompt is obtained?
>
> Here, “valid’’ means that a candidate jailbreak prompt can effectively elicit unsafe content from the target LLM. During optimization, we periodically evaluate the current prompt and terminate the optimization process once it becomes effective. As a result, the actual number of iterations is usually much smaller than 100.
>
>
> **W3**. Regarding the experiments: Why does I-GCG perform worse than GCG? Furthermore, why does LLM-Adaptive perform poorly when attacking Vicuna and Mistral? In its original paper, the success rate against these two models was close to 100%.
>
>
> For fair comparison, all baselines are restricted to a 100-step optimization budget in our evaluation. I-GCG and LLM-Adaptive typically require thousands of iterations to achieve high ASRs (LLM-Adaptive uses up to 10,000 steps in its original paper). In contrast, UJA is much more efficient since it can achieve high ASRs within 100 iterations.
>
>
> **W4**. The writing strongly requires enhancement, including but not limited to streamlining verbose descriptions and ensuring the correct use of mathematical notation.
>
> We thank the reviewer for the suggestion. In the next revision, we will further refinesome descriptions and mathematical notations.

---

> > ### Author Response · Authors · 2025-11-28
> > **Reply to Reviewer rB23**
> >
> > Dear Reviewer rB23,
> >
> > We are grateful for your careful assessment and the valuable feedback you provided. Please let us know if any additional details or clarifications would be helpful on our side. We truly appreciate your contribution to improving our work and look forward to hearing from you again.
> >
> > Best,
> >
> > The Authors

---

### Author Response · Authors · 2025-12-02
**Reply to AC**

Dear AC,

We sincerely thank the reviewers for their valuable comments and suggestions, which have greatly improved the quality of our work. In this revision, we have addressed all major concerns and summarized the key changes for clarity.

Methodology (3)

-	Update the formalization of the logits representation z in Section 3.1. (Page 3, Reviewer ogst W2)

-	Update the description of UJA’s decoding strategy following the COLD-Attack in Section 3.3 (Page 6, Reviewer ogst W2, Reviewer rB23 W2)

Experiments (4)

-	Add the latest baselines and judge models, as well as updated metrics and experimental settings, to Section 4.1. (Pages 6–7, Reviewer rB23 W3, Reviewer ogst Q1-2 and W3, Reviewer ZE6f W3, Reviewer ZE6f W2)

-	Add experimental results and analysis of three baselines (AdvPrompter, ECLIPSE and PAIR) to Tables I in Section 4.2 (Pages 7-8, Reviewer ogst W3 and Reviewer ZE6f W3)

-	Add Qwen3Guard experimental results as an additional judge model to Table 6 in Section 4.4. (Pages 9-10, Reviewer ZE6f Q1 and EaDK Q2)

-	Add experimental data and analysis of the Ablation on ASR metrics to Table 7 in Section 4.4. (Page 10, Reviewer rB23 W1 and Reviewer ogst W1)

Appendix (6)

-	Add an introduction to latest baselines, judge models and harmfulness classifiers in Appendix E-G. (Pages 15-16, Reviewer rB23 W3, Reviewer ogst Q1-2 and W3, Reviewer ZE6f W3, Reviewer ZE6f W2)

-	Add the detailed response generation settings of UJA and baseline to Appendix H. (Page 16, Reviewer rB23 W1 and Reviewer ogst W1).

-	Add the detailed of decoding strategy of UJA to Appendix K. (Page 18, Reviewer ogst W2).


Best,

The Authors

---

### Note · Authors · 2026-01-21

I have read and agree with the venue's withdrawal policy on behalf of myself and my co-authors.